# Volcanoes stunt nearby glaciers

Tryggvi Unnsteinsson [1] ✉, Matteo Spagnolo [2], Brice R. Rea [1], Társilo Girona [3,4], Iestyn Barr [5] & Donal Mullan[6]

Glacierised volcanoes pose significant hazards to societies. Monitoring these volcanoes is therefore essential, though challenging, as traditional geophysical and geochemical methods for tracking volcanic activity can be hindered by glacier cover or remoteness. In this study, we analyse to what extent 307 active volcanoes worldwide impact the mass balance of glaciers within a 40 km radius, a total of 40667 glaciers, by comparing their relative median elevations. We show that glaciers near volcanoes are progressively confined to higher elevations closer to volcanoes, indicating that volcanic processes have a negative impact on nearby glacier mass balance. The influence of noneruptive and eruptive volcanic activity on glacier mass balance is primarily confined to glaciers within 5 km of volcanoes, a similar footprint to detected anomalies in thermal and deformation studies. Our results present a global assessment of volcanic effects on glacier mass balance, and could serve as a baseline for future monitoring efforts of how volcanic activity affects glaciers.

Volcanic eruptions that take place under, or in the vicinity of, glaciers can have devastating consequences for societies and infrastructure[1]. While glacierised volcanoes are challenging to monitor with traditional geophysical and geochemical techniques due to remoteness and glacier cover, observations of volcano-driven glacier melting, importantly, can provide information regarding underlying volcanic activity[2]. For example, localised melt features such as ice cauldrons[3,4] and glaciovolcanic caves and chimneys[5,6] have been commonly observed and used to interpret the underlying volcanic activity. However, in addition to monitoring of such features requiring both high spatial and temporal resolution, they have only been observed within a limited number of glaciers on volcanoes[2]. A promising option to expand observations to more glaciers is the assessment of the distributed effect that volcano-induced melting has on overall glacier mass balance. Observations of changes in glacier mass balance or stark differences between nearby glaciers, that can not be attributed to climate or other forcings, can reveal underlying volcanic activity. Such studies exist but have so far been sparse and spatially limited[2], and have either directly quantified the volcano-driven melt using volcano- and glacier-specific field measurements[7], or inferred it by comparing elevations of debris-free and land-terminating glaciers, ≥1 km[2], within 15 km of volcanoes[8]. As glacier-specific field measurements are scarce

globally[9] and due to the varied characteristics of many glaciers, these methods cannot be used to assess volcanic impacts on all glacierised volcanoes. In a global context, this leaves us with two questions: do volcanoes affect the mass balance of glaciers (which has potential implications for glacierised volcano monitoring) and, if so, how far does this effect extend?

In this study we address the foregoing questions by looking at an easily-measurable proxy for glacier mass balance history: median glacier elevation. Median glacier elevations are comparable to theoretical steady-state equilibrium-line altitudes (ELAs)[10], a common descriptive metric of glacier mass balance[11]. We thus employ a similar methodology to ref. [8], but glacier median elevations are simpler to measure and do not require selection of scaling factors. Encoded in the median glacier elevation are the integrated effects over time of all components of the glacier mass balance: sub-, en-, and supra-glacial ablation and accumulation, along with ice flux. Greater accumulation and/or lesser ablation will drive advance of glaciers downslope, and conversely greater ablation and/or lesser accumulation will force glaciers to retreat to higher elevations. Differences in median elevation between nearby glaciers, where climatic forcing and consequent surface mass balances should be similar, can hence be examined to assess the potential effect of volcanism on glaciers[8,12]. Observed effects on glacier

[1]School of Geosciences, University of Aberdeen, Aberdeen, Scotland, UK. [2]Department of Earth Sciences, University of Torino, Torino, Italy. [3]Geophysical Institute, Alaska Volcano Observatory, University of Alaska Fairbanks, Fairbanks, AK, USA. [4]Geosciences Barcelona (GEO3BCN-CSIC), Barcelona, Spain. [5]Department of Natural Sciences, Manchester Metropolitan University, Manchester, England, UK. [6]School of Natural and Built Environment, Queen's University Belfast, Belfast, NI, UK. ✉e-mail: t.unnsteinsson.23@abdn.ac.uk

elevation and hence mass balance are a snapshot of the long term adjustment of glaciers. The response time of glaciers to a mass balance forcing, e.g. volcanism, will depend on the scale of the forcing, climate, and glacier size, and could span multiple decades or longer[13]. Thus, any volcanic effect that we observe on glaciers will be an integrated effect of all prior volcanic processes and events –such as volcanic eruptions, interactions with volcanic products, and volcanically-enhanced basal melting. Here, we analyse the relative difference in median elevations of all glaciers within a 5, 10, 20, 40 km radius of the location of each of the world's active Holocene volcanoes (erupted in the past 12000 years[14]). We do this by utilising global datasets of glaciers and volcanoes: the 274531 glaciers in the Randolph Glacier Inventory (RGI)[15] and the 1309 Holocene volcanoes from the database of the Smithsonian Institution's Global Volcanism Program (GVP)[14]; thereof are a total of 40667 glaciers located within 40 km of 307 active volcanoes.

## Results and discussion

### Detection of anomalous median glacier elevations

Our comparison of median elevations of nearby glaciers (Methods) shows evident various regional and local trends along with anomalies in relative glacier median elevations. Gradual variations in relative median glacier elevations are generally observed across individual mountain ranges and other glacierised regions. Extensively glacierised areas have more uniform relative median glacier elevations than areas with smaller glaciers. For example, the Alaska Range shows higher regional uniformity than the Coast Range of British Columbia. As expected[13], these variabilities are most often evidence of local accumulation and ablation gradients, e.g., with relative median elevations increasing with distance from the nearest moisture source or more solar-facing aspects. Regardless of the general trend within any region, there are some notable exceptions where individual glaciers have significant, anomalous, relative median elevations. Such anomalies are mainly associated with three types of glaciers: calving glaciers; debris covered glaciers; and glaciers that may have been inaccurately mapped or mapped as separate disconnected glaciers in the RGI dataset. Dry and wet calving glaciers both have positive relative median elevation anomalies, i.e., their median elevation is higher than that of surrounding glaciers since calving can present a significant, negative, non-surface mass-balance term (e.g., small dark red glaciers in Figs. 1b, S1, and S4). Debris covered glaciers commonly show a negative anomaly, likely due to reduced ablation (e.g., dark blue glaciers in Figs. 1b, S2, and S4). We expect such anomalies as they are a manifestation of additional forcing on glacier mass balance that are specific to each

glacier, and not related to the regional climate control. However, an additional anomaly can be found throughout the dataset, which is analysed in detail in the following section.

### Volcano induced glacier mass-balance anomalies

Our results indicate that there are consistent positive anomalies in median glacier elevations near the world's volcanoes. From an analysis of 40667 glaciers located around volcanoes, our results show that median glacier elevations increase with proximity to volcanoes (Methods) for the majority, up to 80%, of glacierised volcanoes active at some time during the Holocene (Table 1 and Figs. 2 and 3). That is, glaciers closer to volcanoes are progressively confined to higher elevations. This trend is evident for all search radii used in this study (5, 10, 20, 40 km), with the strongest signal for the smaller radii (Table 1). The median elevations of glaciers closer to volcanoes appear as positive anomalies, rising at an average rate of around $55 \pm 5$ m km$^{-1}$ ($p \ll 0.01$) towards volcanoes (Fig. 4). For statistical significance, we limited the analysis to volcanoes with at least four nearby or overlying glaciers. Despite not filtering out, or adjusting for, different glacier types or characteristics, the positive anomalies are still observed globally in all statistical tests (Table 1 and Methods). The Mann-Kendall test shows weaker correlation than the other tests for larger radii (Table 1), which is expected as it does not directly consider the distance from the volcanoes (Methods). The considerable agreement between the tests demonstrates that the signal is strong enough to offset the potential effect of other factors that are known to affect glacier mass balance[13], e.g., aspect and distance from moisture source, which might be able to mask the impact that volcanoes have on glaciers. Further, we observe this trend for various sub-types of volcanoes, irrespective of their eruptive histories or elapsed time from their last eruptions.

There are likely three main controlling factors pertaining to volcanoes that could influence the relative elevation of nearby and overlying glaciers: presence and production of debris; prominence above surrounding topography; and volcanic heat fluxes. Volcanoes generally produce high fluxes of sediments from their flanks[16], and can be subject to heightened levels of erosion by the activity of overlying glaciers[17]. This, in conjunction with volcanic ejecta, may enhance debris entrainment and covering of glaciers proximal, or down wind of, volcanoes. As debris covers thicken they rapidly begin to shield glaciers from ablation[18,19], resulting in lower median glacier elevations. A visual inspection of our results confirms that thicker and more expansive debris coverage is indeed linked to lower median elevations of glaciers both near and away from volcanoes (e.g., dark blue glaciers

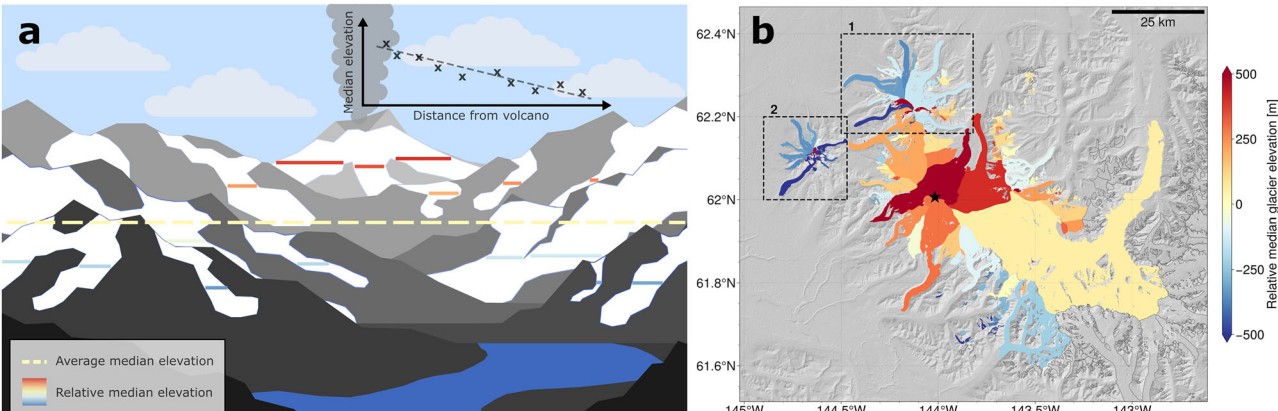

**Fig. 1 | Schematic and example of higher median glacier elevations near volcanoes. a** An illustration demonstrating how median glacier elevations (solid lines and **x**'s in inset graph) rise towards volcanoes. The colour gradient of the median elevations indicates the relative median glacier elevations, computed as the deviation from the mean (dashed yellow line): relative median glacier elevations closer to volcanoes are positive (red), but trend to negative (blue) further away.

**b** An example of median glacier elevations rising towards Mount Wrangell volcano (black star), Alaska. The glaciers within 40 km of Mount Wrangell are coloured according to their relative median elevation. The dashed boxes are to highlight the debris-covered glaciers and dry-calving glaciers of Sanford (**1**) and Drum (**2**). The base map is a shaded relief of the ASTER GDEM V003[38].

in Figs. 1b, S2, and S4). Hence, it seems improbable that debris-shielding is driving higher median elevations around volcanoes. Conversely, impurities in ice or sufficiently thin debris cover, e.g., tephra from eruptions, can enhance ablation by decreasing glacier surface albedo and increasing solar energy absorption[20,21]. This could result in higher median glacier elevations. Debris-enhanced melt could be particularly prevalent following volcanic eruptions that blanket glaciers with tephra[21]. However, glaciers can also be positively affected if the tephra layer is thick enough[20,22]. The longer-term effect of erupted tephra on glacier ablation is complex and site specific. In the years immediately following an eruption, tephra may both: continue lowering the albedo of glaciers, potentially increasing ablation, before rebounding to previous levels[21,23]; or carry on shielding and reducing ablation rates, leading to glacier preservation and potential advance[24,25]. Subsequently, reworking and redistribution of tephra on glaciers can also both promote and inhibit ablation depending on glacier and surface characteristics[25,26]. Thus, due to the potentially diverging effects tephra can have on glaciers, tephra-enhanced melting is unlikely to alone explain the global signal of higher median glacier elevations. This is further supported, as only around 30–35% of glacierised volcanoes had confirmed eruptions in the last century[14].

Volcanoes that build edifices above the local terrain reach higher altitudes and therefore experience lower temperatures, which can increase the potential for glacier growth and survival[13]. This raises the question of whether higher median glacier elevations near volcanoes are simply a by-product of the topographic prominence of volcanoes. However, in the absence of other influencing factors, these higher-reaching accumulation zones on volcanoes should yield lower minimum glacier elevations as more accumulation translates to higher ice flux downslope. Hence, glaciers on, or near, towering volcanoes should be longer and have comparable median elevations to glaciers sitting in lower neighbouring non-volcanic terrain. This is the case in our results for glaciers near other non-volcanic prominent peaks around the world, e.g., in the Alaska Range, St Elias Mountains, Alps, and High-mountain Asia (Supplementary Figs. S1, S2, S3, and S4). Instead, for glaciers near and overlying volcanoes we observe that as relative maximum glacier elevations increase nearer to volcanoes, the relative minimum glacier elevations stay relatively constant (Fig. 4). Whereas theory would suggest that higher maximum elevations should correlate with lower minimum elevations. That is, glaciers on or near volcanoes have a reduced elevation range, and hence shorter length than expected for glaciers sitting on topographically prominent peaks. Consequently, relative median glacier elevations rise towards volcanoes.

Given the caveats associated with the two other main drivers discussed above, the most plausible driver of higher median elevations for glaciers located on or around volcanoes, in a global context, would seem to be volcanically-induced basal melt. This melt could be attributed to any volcanic process during both quiescent and active periods, and either be localised or distributed. Increased basal melting will by itself act to shrink glaciers[7], but it may also influence glacier velocities by enhancing sliding[27]. Short periods of increased velocities could elongate glaciers but, in the long term, sustained faster velocities would likely yield an accelerated ice flux, and thus negatively impact glacier mass balance leading to glacier shrinking and retreat to higher elevations.

Our results demonstrate a volcanic effect on glacier mass balance, through a rise in median glacier elevations, around the majority of volcanoes globally. This suggests that the number of volcanically affected glaciers could be in the thousands (Table 1), while previously only about 150 "volcanically affected" glaciers had been documented worldwide[2]. This under-reporting is likely due to the reliance on direct observations of localised melt features, whereas modern remote-sensing techniques are now revealing the true extent of volcanic effects on glaciers[27]. The mismatch between direct observations and global assessments of volcanic effects on glaciers suggests that melt processes with no obvious surface expression and associated ice dynamic adjustments, which are difficult to directly observe, play a significant role in glacier mass balance. That is, the main driver of elevated median glacier elevations near volcanoes globally must be volcanically induced melting, and any resulting dynamic responses. This volcanic effect may result from any, or combination, of aforementioned volcanically-induced melt processes, i.e., enhanced basal melting and, where applicable, tephra-enhanced melt.

**Table 1 | The total number, $N$, of volcanoes and glaciers within a given radius and the percentages, $P$, of all the volcanoes, with at least four glaciers, that show an increase in median glacier elevations towards them**

| Radius [km] | $N_{volcanoes}$ | $N_{glaciers}$ | $P_{LR}$ [%] | $P_{MK}$ [%] | $P_{SR}$ [%] |
|---|---|---|---|---|---|
| 5 | 214 | 3290 | 77.4 | 80.8 | 79.4 |
| 10 | 228 | 6214 | 79.9 | 78.3 | 81.3 |
| 20 | 260 | 14366 | 79.3 | 74.0 | 79.7 |
| 40 | 307 | 40667 | 71.5 | 60.2 | 74.5 |

Three statistical tests (see Methods) are used to determine the trend of median glacier elevations around volcanoes: linear regression (LR), Mann-Kendall test (MK), and Spearman's Rho test (SR).

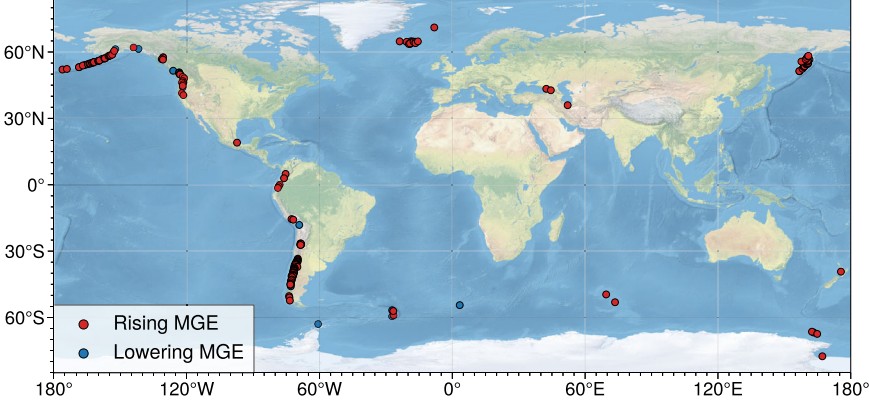

**Fig. 2 | Global prevalence of higher median glacier elevations around volcanoes.** The local trend of median glacier elevations (MGE) for 5 km around glacierised volcanoes. Red and blue dots denote rising and lowering, respectively, median glacier elevations towards volcanoes based on the Spearman's Rho test. Up to 80% of Earth's volcanoes demonstrate higher median glacier elevations closer to volcanoes (Table 1). The basemap was made with Natural Earth (naturalearthdata.com).

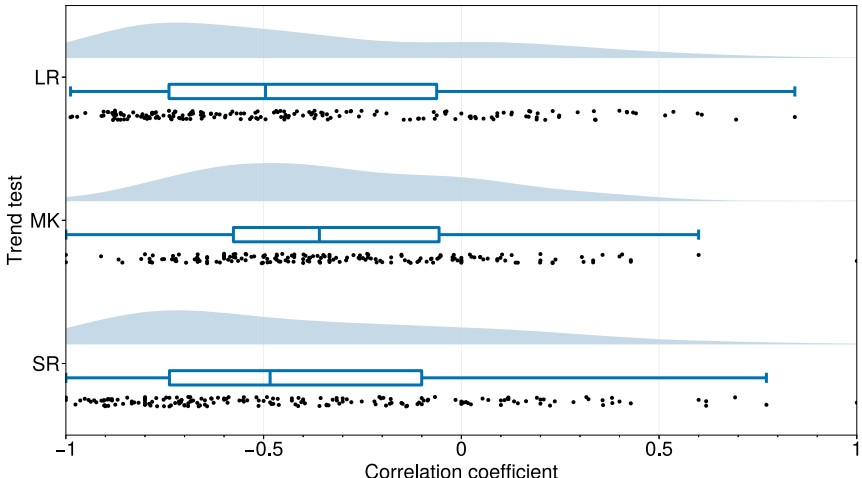

**Fig. 3 | Trend analysis of median glacier elevations surrounding volcanoes.** The correlation coefficients of different tests of the local trends of median glacier elevations for 5 km around each glacierised volcano, shown as individual values (black dots), median and quartile values (dark blue box plots), and value distribution (light blue ridge plots). The trend tests are: linear regression (LR), Mann-Kendall trend test (MK), and Spearman's Rho test (SR). A correlation coefficient of 0 means no trend, ±1 indicates a perfect linear and/or monotonic trend, and the sign denotes the direction of the trend away from volcanoes. The trend tests show a definite negative correlation between median glacier elevations and the distance from volcanoes, i.e., median glacier elevations generally fall moving away from volcanoes.

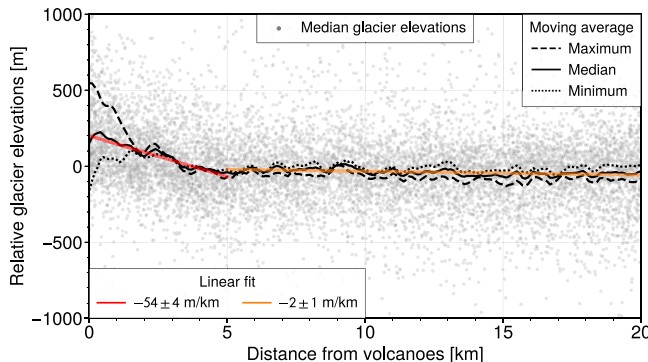

**Fig. 4 | The trend of all median elevations of all glaciers around all volcanoes.** Relative median glacier elevations for all glaciers (gray dots) within 20 km of all volcanoes, a total of 14366 glaciers. The median glacier elevations are shown as relative to the average within the radial search areas. To highlight the general trend, a smoothed moving average (black lines) is shown of the relative minimum- (dotted line), median- (solid line), and maximum- (dashed line) glacier elevations. A clear rise in median glacier elevations can be seen towards volcanoes within a distance of 5 km (red solid line), but beyond 5 km there is no trend (red dashed line).

Although the goal of this study is global and not focused on the analysis of individual volcanoes, it is important to drill a little deeper on some of the outliers. Two main reasons for glacier-on-volcano outliers are observed: (i) domination of non-volcanic factors on glacier mass balance; and (ii) poorly constrained volcano locations. For example, on Aniakchak, Alaska, there are small glaciers situated on either side of its southern caldera rim. The median elevations of its glaciers are clearly controlled by aspect, with south-facing glaciers (outside the caldera) being smaller and lying higher than those that are north-facing (inside the caldera). Likewise, several sub-Antarctic island-volcanoes show negative median glacier elevation anomalies or otherwise ambiguous trends. For some, their analysis may be dominated by a disproportionately high fraction of glaciers calving into the ocean compared to other glacierised volcanoes. Several subantarctic volcanoes also have poorly constrained locations in the GVP database[14], e.g., Deception- and Candlemas Islands which have locations off-island in the ocean. Some outliers could likely be dealt with by controlling for non-volcanic factors that could influence glacier elevation, or by adjusting volcano locations. However, as such corrections would have to be applied volcano by volcano and potentially, glacier by glacier, to the entire datasets, we choose to not impose any biases or culling of the data. The presence of a strong global signal, despite these outliers, is a good indication of the robustness of our results.

## Footprint of volcano-induced glacier median elevation anomalies

The lateral extent, i.e., the footprint, of detectable signs of volcanic unrest varies for different ground and remote sensing observations. Ground deformation prior to volcanic eruptions is most likely to be detected within 5 km of volcanoes[28], and the expected area of small scale thermal unrest may be even smaller[29]. These observations could further be spatially limited by glacier cover, as snow and ice can mask out thermal-[30] and deformation signals[31]. We carried out the analysis of median glacier elevations around volcanoes for search radii of 5, 10, 20, 40 km to investigate how far the volcanoes could affect the glaciers. While an overall trend of increasing median elevation with proximity to volcanoes is present for all search radii, the stronger signal is found within the first 5 km. Although this may vary between individual volcanoes, aggregating all glaciers near volcanoes indicates that volcanoes mainly influence the mass balance of glaciers that lie within 5 km radius (Fig. 4). Elevated volcanically-induced melt rates are likely thus principally confined to volcanic edifices. This is in accordance with the identification of anomalies in the aforementioned deformation and thermal studies of volcanoes. Our results highlight where future long-term monitoring efforts of glacierised volcanoes, using glacier mass balance to infer on volcanic activity, should be focussed i.e. within a 5 km radius.

## Global implications

We have demonstrated that, globally, volcanoes have a negative impact on the mass balance of overlying and nearby glaciers, i.e., median elevations of glaciers are higher the closer they are to volcanoes, especially within a 5 km radius. These negative volcanic effects on glacier mass balance are driven by localised and/or distributed melt processes. Since observations of localised melt features are relatively few, we deduce that the most significant effect, globally, is distributed

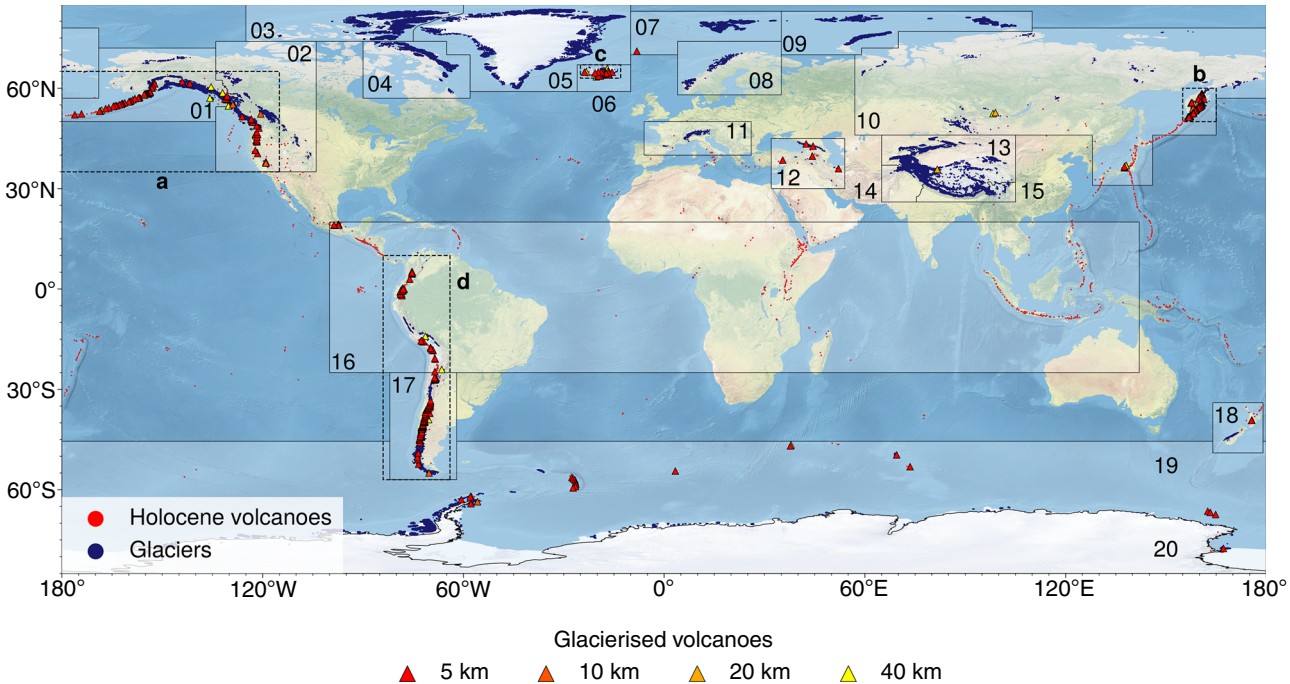

**Fig. 5 | Global distribution of glaciers and volcanoes, highlighting glacierised volcanoes.** Holocene volcanoes (red dots) from the Global Volcanism Program[14], and glaciers (blue polygons) and first-order regions (black polygons) from the Randolph Glacier Inventory[15]. The glacierised volcanoes analysed in this study (triangles) are volcanoes with glaciers within a distance of 5 km (red), 10 km (dark orange), 20 km (light orange), and 40 km (yellow). The basemap was made with Natural Earth (naturalearthdata.com). Antarctic volcanoes, i.e., volcanoes within RGI region 20, are here excluded from the list of glacierised volcanoes. Enlarged insets of the areas with the highest concentration of glacierised volcanoes (dashed boxes **a**–**d**) are shown in Fig. S5.

volcanically-induced melting and resulting dynamic feedbacks. We have further shown that this volcano-induced mass-balance anomaly can be observed without the need for in-situ glacier-specific mass and energy balance studies. In a warming climate, this additional negative effect that volcanoes have on the mass balance of overlying glaciers could hasten their demise. Most importantly, our results present a reliable baseline glaciological metric to be used to investigate how glaciers are affected by volcanic activity. This could potentially aid future monitoring efforts of glacierised volcanoes using glacier mass balance as a proxy for volcanic activity.

## Methods
We locate glaciers within the vicinity of volcanoes by comparing two databases: the "Volcanoes of the World" from the Smithsonian Global Volcanism Program (GVP)[14]–a complete list of Earth's Holocene volcanoes; and the Randolph Glacier Inventory (RGI) version 7.0[15]–a dataset containing outlines and information for all glaciers in the World, excluding the ice sheets of Greenland and Antarctica, around the target year of 2000 (Fig. 5). Combining the GVP and RGI databases to locate glacierised volcanoes has previously been done by refs. 1,32, who used a circular buffer, of radius 5 km, to determine whether a volcano is glacierised or not. We follow the same methodology as[1,32] but provide automated Python scripts which allow the user to specify the search radius. The scripts rely on the Python package `GeoPandas`[33] to manipulate and analyse all geospatial data. The scripts first separate the volcanoes into the RGI regions (Fig. 5), and then iteratively find all glaciers within a given radial distance for all the regional volcanoes. Both datasets use the WGS84 coordinate system, but the radial search area is created in each volcano's respective UTM coordinate system, found using `GeoPandas`' inbuilt function `estimate_utm_crs`, and then converted into WGS84. The scripts output whole glacier geometries, i.e., the full geometries of glaciers that partially intersect with the radial search area are returned. The topographic attributes of

glaciers used in this study are already included in the RGI v7.0 dataset[15]. These topographic attributes were computed using the Copernicus DEM[34] for all but a few glaciers[15,35] where the alternative DEMs were used: RAMP[36], DEM3[37], ASTER[38], or TANDEM[39].

This global study analyses if, and how, glacier median elevation, a metric for glacier mass balance, changes with proximity to volcanoes. To do so, we look at trends in glacier median elevation for each volcano that, for statistical significance, has at least four glaciers within the specified search radius. Reference [8] looked at the ELA of glaciers surrounding several volcanoes in South America. We build on the methodology of ref. 8, adapting it to use median glacier elevations, $\tilde{z}$, instead of ELAs, and analysing them as a continuous function of distance rather than discrete intervals. As a descriptive distance between volcanoes and glaciers, we use the minimum distance between volcano locations, given in ref. 14, and glacier outlines, with zero indicating that a volcano is directly overlain by a glacier. We compute the relative median elevation for each ($i$-th) individual glacier around a given volcano as

$$\Delta \tilde{z}_i = \tilde{z}_i - \bar{\tilde{z}}, \tag{1}$$

where

$$\bar{\tilde{z}} = \frac{1}{n} \sum_{i=1}^{n} \tilde{z}_i \tag{2}$$

is the average median glacier elevation within the radial search area, comprising of $n$ glaciers. In order to investigate anomalies outside of volcanic areas we similarly analyse median glacier elevations around all 274531 glaciers in the RGI dataset[15]. Instead of the circular search areas around volcanoes, for glaciers we use buffered geometries of the glaciers themselves as search areas. These buffered geometries do approach circular geometries when the search radii become larger

than the dimensions of glaciers. We hence collectively refer to all search areas around volcanoes and glaciers as "radial search areas". The analysis of median glacier elevations within radial search areas around glaciers is otherwise identical to that of around volcanoes.

Investigating the influence of different physical parameters on global, and volcanic, glacier elevations, requires fitting the observed data to models. We rely on simple statistical models to analyse the influence of different glaciovolcanic parameters but leave detailed numerical investigations for future work. We fit these models to the glaciers surrounding each volcano, excluding only glaciers closer to another volcano. The simplest model that we use to identify parameter relations is (simple-) linear regression. We use the Python packages `SciPy`[40] and `statsmodels`[41] to carry out the linear regression, fitting the regression coefficients to variables using the ordinary least squares method. The quality of the regression is captured by the Pearson Correlation Coefficient, $-1 \leq r \leq 1$, with -1 and 1 denoting perfect negative and positive correlations, respectively. We further use the Mann-Kendall trend test (`pyMannKendall`[42]) and the Spearman's Rho test (`Scipy`[40]) to estimate the degree of monotonic trends of relative median glacier elevations around volcanoes. The Mann-Kendall test and the Spearman's test give statistics $-1 \leq \tau \leq 1$ and $-1 \leq \rho \leq 1$, respectively, both demonstrating how monotonic a trend is, with $\pm 1$ indicating a perfectly monotonic trend and the sign denoting negative or positive[43]. The null hypotheses for the regressions and the trend tests, is that there is no trend or correlation to be found.

Our study relies on the RGI[15] and GVP[14] open datasets and hence inherits any errors and uncertainties present in those datasets. We do not correct for any presumed erroneous volcano locations or glacier geometries, carrying out our analysis with both datasets "as is". For the uncertainties of each dataset, we refer readers to their respective references, as well as the Open Global Glacier Model[35] which was used to compute the elevation attributes of glaciers within the RGI dataset. We assume that the uncertainties, and/or errors, for both datasets are random (and none have been reported elsewhere) and will hence not systematically affect our results. Glacier elevation trends vary considerably between individual volcanoes, and we do not attempt to model the trends explicitly. However, the linear model gives a representative trend around all volcanoes, along with the directionality provided by the other trend tests. The Mann-Kendall test is expected to underperform the Spearman's Rho test as it only considers ordered data, i.e., as if all glaciers were evenly distributed away from a volcano. Due to the limited number of glaciers surrounding many volcanoes, we recognise that *p*-values of the regression and trend tests may be elevated for each individual volcano. However, the aggregation of data and trends for the whole global volcano population reduces this ambiguity and serves as a testament to the meaningfulness of individual regressions and trend tests (Fig. 4).

## Data availability
All data used in this study are open access and available from the sources referenced. The global glacier outlines and attributes are from version 7.0 of the Randolph Glacier Inventory (RGI)[15], and the volcano locations and attributes are from the Smithsonian Institution's Global Volcanism Program (GVP)[14]. The results from the trend analysis for each volcano are provided with the code (see below).

## Code availability
All the code developed for this study to analyse the RGI and GVP datasets is built upon open-source Python packages, and is available at: https://github.com/TryggviU/glac_by_volc[44].

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

## Acknowledgements

T.U. was supported by UKRI QUADRAT Doctoral Training Partnership studentship (NE/S007377/1). T.G. and T.U. gratefully acknowledge funding from the U.S. Geological Survey under Cooperative Agreement No. G24AC00240. Valuable discussions with colleagues at the Alaska Volcano Observatory are also acknowledged.

## Author contributions

T.U., M.S., B.R.R., T.G., I.B., and D.M. conceptualised and designed the study. T.U. led code development, data analysis, interpretation, and manuscript writing. M.S., B.R.R., T.G., I.B., and D.M. provided supervision, and contributed to data analysis, interpretation, and manuscript writing.

## Competing interests

The authors declare no competing interests.
