## [Transparent Peer Review file · Nature Communications]

Volcanoes stunt nearby glaciers

Corresponding Author: Mr Tryggvi Unnsteinsson

Version 0:

Reviewer comments:

Reviewer #1

(Remarks to the Author)

Review of Unnsteinsson et al., "Feeling the heat: globally, volcanoes stunt nearby glaciers" Comments by Linda Sobolewski

This manuscript presents a very interesting study in which the authors analyze to what extent active volcanoes impact the mass balance of adjacent glaciers by looking at their relative median elevations in comparison to non-volcanically influenced glaciers. I think it is a very interesting approach to assess the global influence of volcanoes on glaciers apart from disruption by major (single) events. The methods and data used in this study are comprehensible and well presented. Furthermore, the strengths and weaknesses of this work were identified and discussed.

The key results of this study are that volcanoes affect the mass balance of glaciers globally, even without having a major eruption. Relative median elevations increase with proximity to volcanoes (the closer the stronger the effect) with volcanic heat fluxes being identified as the main driving factor. Other effects were also taken into account and exceptions were presented. While I started reading the manuscript some questions were coming up, e.g.: How to exclude the effect of climate? Considering small scale characteristics and regional specifics? However, all these questions were answered in detail in a clearly understandable way and context was provided where necessary. Overall, I think this work presents important and also very interesting (and in the end surprising) results to better understand volcano-ice interactions in a global context. Although it is known that volcanoes often impact overlying or nearby glaciers—even in periods of quiescence, the percentage (~80%) is surprising.

The literature used in the manuscript is appropriate and includes very recent studies.

Although I think the work presented is very good and the manuscript is well written, I have some suggestions for improvement and I would like to see some changes of figures. After making these minor revisions I would recommend the manuscript for publication.

Some specific comments are:

Lines 45-56: I understand that the authors somehow have to provide an overview of what they have done and some context is needed. However, this part really reads like a methodology section. Is there a way to summarize this paragraph in a more general way and move the details to the methods section?

Section 2.4: I would consider having the "Conclusion" as a separate chapter since the caption for chapter 2 is "Results and discussion".

One general comment regarding the figures: I know it's not always easy to calculate the final size of figures in a manuscript, but I would try to roughly use the same size for legends and labeling of graphs. Sometimes the labels are very small and hard to read, even when zooming in. And sometimes they are disproportionately big compared to the actual figures.

Figure 1a: I really like the concept of this one but it took a while before I recognized the blue lines in this figure (below the yellow one). Is there a possibility to play a bit with the colors and the contrast? Maybe make the background colors a bit lighter or more transparent to better highlight the relative median elevation lines? If the "Median elevation" belongs to the inset graph (upper part of the figure) shouldn't it be dashed?

Figure 1b: Please make the labels a bit bigger.

Figure 2: This might be a subjective opinion but would it be better not to make the dots transparent? It appears that you used more colors than red and blue since sometimes the dots look more orange and sometimes really red. I don't think this will influence the key message of your figure although not every single dot will be visible. OR: did you use more colors to show a wider range? Then you should use a different color scale such as in Fig. 1a. Also, please make the legend a bit smaller and not hide data with it.

Were there any volcanoes where the MGE wasn't influenced at all?

Figure 4: It's super hard to distinguish between the different lines. Would it be possible to work with different colors or would it significantly influence the message of your figure when you zoom a bit more to the center and not show the full extent towards the upper and lower values? Or perhaps a 4a and 4b version where 4b is the version zoomed in to a scale that allows the reader to visualize the differences in the lines.

Figure 5: I really understand the difficulty plotting so much data on one single map. Maybe plot individual areas separately to better visualize details? See for example Sobolewski et al. (2022), Figure 1 (<https://doi.org/10.1007/s00445-022-01525-z>)

(Remarks on code availability)

The author provided a README file with very detailed instructions of how to run the code and reproduce his results. All data sets needed for this are accessible. Apart from providing the code the author also included figures and tables which makes it possible to have a closer look at individual volcanoes/glaciers and related results. Overall, the approach is comprehensible and reproducible.

Reviewer #2

(Remarks to the Author)

Review of Unnsteinsson et al., NatComm, March 2025

In their paper, Unnsteinsson et al. build on an earlier study by Howcutt et al (Geology, 2023, restricted to the Andes) to detect the influence that volcanic activity has at global scale on glacier health. This is done by intersecting two main database: a global inventory of glaciers and of actives volcanoes. The authors use the anomalies of the median glacier elevation (compared to a regional mean) to detect if volcanoes influence the glacier mass balance. A higher median glacier elevation is indicative of a shorter glacier and hence one for which there is an additional source of melt, not climate related. Authors conclude to a clear imprint of volcanic activities on glacier health.

This is an exciting study that, if this is confirmed, suggests a distinct influence of volcanic activity on glacier mass balance. I think it is appropriate for Nature Communication given the length and the fact that it elegantly intersects two fields of research, glaciology and volcanology. However, as explained below I am less convinced by the demonstration that this purely due to enhanced basal melting and also a proper definition of the criteria used to qualify "glacier health". "Mass balance" is maybe not the right term, maybe even wrong...

General comments.

1/ Ref [9] cited in the introduction is a global dataset of individual glacier mass balance spanning the period 2000 to 2019 (sub-periods are also available but the longest period has the smallest uncertainties). Rather than relying on an indirect proxy of the mass balance (the median altitude, as also done in Howcutt et al.), authors could apply their methodology to this dataset. In my view, that could be an improvement of the paper and a strong confirmation that glaciers are actually feeling the heat of volcanoes.

This is a general comment I wrote right away after reading the paper a first time and in particular the abstract where the word "glacier mass balance" is used several time.

Thinking about it, maybe I missed something and maybe the dataset in Ref [9] (i.e. their 20-yr glacier mass balance) is not the right one to detect the influence of volcanoes on glaciers. A reason for that would be that glaciers influenced by volcanoes already had time to build an hypsometry, different from their non volcanic neighbours (i.e. a different median altitude), such that the enhanced volcanic melt is "already" compensated by a location at higher reaches (where surface mass balance is more positive due to the mass balance gradient). If this is indeed the case, then the dataset in Ref [9] may not really help to detect the influence of volcanoes (backing up the choice by the authors). These issues of response time, hypsometric adjustment and hence of the choice of the right variable to qualify the impact that volcanic activities have on glacier health is not discussed at all in the submitted manuscript and this is really missing. In fact, if my reasoning above is true, it means that glaciers on volcanoes do NOT have mass balances different from others, but rather that they built an hypsometry that takes into account this additional source of melt. So the word "mass balance" may not be the right one to use.

2/ Authors could also discuss at what time scale their results apply. Echoing a bit my first main comment just above, I guess this time scale is the time for the glacier to built an hypsometry where the volcanic effect is compensated by the distribution with altitude. An interesting follow up question is how glaciers close/far from volcanoes will respond to ongoing warming. Do authors expect a similar or different response (assuming a similar climate forcings)? This should be touch upon in the article (not solved). A related question is the time-invariance of the volcanic activity. Again something to discuss, even briefly.

3/ "though debris-enhanced melt may play a role for some volcanoes, it cannot explain the global scale of positive median elevation anomalies associated with volcanoes". This is a very firm statement based on little evidences. I would be ready to accept such a statement if, for example, authors had examined some albedo maps during the months of minimum snow cover (from Modis) and confirm that the albedo is not statistically different for glaciers nearby volcanoes. The same applies for debris thickness for which global map of debris thickness could be used to verify that glaciers on volcanoes do not differ (see Herreid and Pellicciotti, 2020; Rounce et al., 2021). I understand this is a significant amount of work but otherwise the conclusion of "no debris influence" is not supported.

Specific comments (including some substantial comments).

L19. It is a bit hard to grasp the effect said this way. "lead to more negative mass balances" would be clearer I think. My main comment 1 above may lead to a rewriting of this and not using "mass balance". The mass balance of volcanoes-influenced glaciers may NOT be different now because glaciers already had time to adjust their hypsometry to difference in melt.

L20 Are the "adjacent glaciers" mentioned above within this distance of the volcanos? (I guess not but the abstract need to define what an "adjacent glacier" is)

L45-56. Authors need to refer to the methods section in the "main"

L49. Radii around what center? Volcanoes? Make it clear and tell the readers how the volcanoes location is obtained. How two nearby volcanoes are treated (to avoid redundancy), as many volcanoes will be clustered. This comment will be easy to answer by referring more specifically to the Methods section (I did not know this Methods existed when I read the article first)

L51. define the related time period (in years)

L56. Median altitude is an indirect proxy of the mass balance and rather reflect the mass balance history. This needs to be clarified (see general comment 1).

Figure 1a. the blueish colours for the median elevations are hard to distinguished from the background.

L68. Authors could annotate panel 1b to make sure this is clear to all readers which glaciers are concerned.

L82. I did not understand this statement. The first part of the sentence is about number of glaciers in the vicinity and the second part of about glacier types.

Table 1. Are the percentages given for the number of volcanoes for which the increase in median glacier elevation is statistically significant? Or for all, i.e. without considering the statistical significance? My understanding is that the authors compared three different ways to compute the trend but when did they accounted for the significance of the trend? Maybe I missed something but this table needs to be clarified.

L94. I find it strange to cite an article in the "Arabian Journal of Geosciences" to support the impact of debris on glacier melt. I can think of this study, looking in detail at the albedo in Iceland but their must be many others
Gunnarsson, A., Gardarsson, S. M., Pálsson, F., Jóhannesson, T., and Sveinsson, Ó. G. B.: Annual and inter-annual variability and trends of albedo of Icelandic glaciers, *The Cryosphere*, 15, 547–570, <https://doi.org/10.5194/tc-15-547-2021>, 2021.

L94. Herreid and Pellicciotti provide a global debris thickness map on glacier that authors could use to test the hypothesis (see general comment).

L98. The statement of few tephra-generating eruptions in the last century needs to be backed up by a reference.

L106. Authors need to show some plots/statistics for Alaska Range, St Elias Mountains, Alps, and High-mountain Asia (in a supplement?) in order to back up the comparable median glacier elevations whatever the maximum height of the summit. Really important.

L112. "The only plausible driver of higher median elevations for glaciers located on or around volcanoes...". "Only" can be used if the effect of debris (albedo) can be ruled out. This is has not been demonstrated yet (see general comments).

Figure 3. Do authors have an explanation why the correlation coefficient seems to agree rather well between LR and SR, but MK seems to lead to less negative values? This is discussed a bit in the Methods section but could be also briefly mentioned in the main.

L125. I did not understand why an enhanced mass turnover would negatively impact the glacier mass balance. Can the authors cite a reference to back up this statement and provide more explanations? Use the term mass balance carefully...

L180. Make it clear that you use the RGI attributes directly and did not recompute them. (if I understood well)

L189. Authors need to clarify how the distance is computed. Do they find the distance to the centroid of the polygon? Or the distance to the closest segment of the polygon?

References

Herreid, S. and Pellicciotti, F.: The state of rock debris covering Earth's glaciers, *Nature Geoscience*, <https://doi.org/10.1038/s41561-020-0615-0>, 2020.

Rounce, D. R., Hock, R., McNabb, R. W., Millan, R., Sommer, C., Braun, M. H., Malz, P., Maussion, F., Mougnot, J., Seehaus, T. C., and Shean, D. E.: Distributed Global Debris Thickness Estimates Reveal Debris Significantly Impacts Glacier Mass Balance, *Geophysical Research Letters*, 48, e2020GL091311, <https://doi.org/10.1029/2020GL091311>, 2021.

(Remarks on code availability)

Version 1:

Reviewer comments:

Reviewer #1

(Remarks to the Author)

Re-review of Unnsteinsson et al., "Feeling the heat: globally, volcanoes stunt nearby glaciers" Comments by Linda Sobolewski

The authors have invested a lot of work to revise the manuscript and address all comments and concerns. I think it is appropriate for *Nature Communication* in its current form and I would recommend the manuscript for publication.

(Remarks on code availability)

Reviewer #2

(Remarks to the Author)

Review of the revised version of Unnsteinsson et al., *NatComm*, June 2025

The article has been improved, taking into account most comments from the original reviews. However, I cannot recommend publication yet because there is still some ambiguity in terminology: median glacier elevation (studied here) and mass balance need to be more clearly distinguished. I tried to highlight all places where some improvements are needed.

Abstract: the background justification of the work is not appropriate. As if an analysis of the impact of volcanoes on glacier health could be a way to "monitor" volcanoes. This would be really indirect because of many other processes influencing glaciers (including climate change) and because the time (at least decadal) it takes for a glacier to adjust its hypsometry to a change in volcanic heat. Hence, the first sentences of the abstract need to be rewritten.

L37-38 unclear what is meant. "local to" seems isolated. "metrics" is vague.

L45. "Median glacier elevations are comparable to equilibrium-line altitudes (ELAs), a common descriptive metric of glacier mass balance"

Need to specify the time scale at which this applied. For example inter-annual variability of the glacier median elevation is very small (glacier geometries change slowly) whereas the mass balance inter-annual variability is large (because of the climate variability).

So this statement only applies to long time scales (century long time scale) and this needs to be explained.

Title of section 2.1. Authors did not detect "mass balance anomalies". But different median glacier elevations. Hence, this section title is misleading.

L93-94. See for example the factors listed in Huss et al., 2012 : <https://tc.copernicus.org/articles/6/713/2012/>

Fig4. I suggest using another color than red for the dashed fit to really make the curves different (hard to see the dashed one as it overlays a black line)

L127-128. Maybe I missed something but there seems to be a contradiction with the previous sentence. If the max glacier elevation increase and the minimum elevation stay constant then the range increase and the length also.

Title 2.3 replace "mass balance" by "median elevation"

L174. I do not think mass balance can be a proxy of volcanic activity. As said in my original review and repeated here, the RGI glaciers already adjusted their hypsometry (as actually demonstrated by this study) to the influence of volcanoes by standing at higher median elevations. Hence, their present-day mass balance is very likely not different to their neighbours.

Fig S1-S4. The number for the summits are so clustered that there are hard to distinguished. Sometime the numbers (in black) are also hard to read because of the dark background colors. These figures need some edits.

(Remarks on code availability)

Version 2:

Reviewer comments:

Reviewer #2

(Remarks to the Author)

Re-Review of the revised version of Unnsteinsson et al., NatComm, July 2025

I would like first to thank the authors for considering my suggestions.

I am not entirely satisfied by their answers but will not argue further here. Their work is extremely valuable and convincing but I am not convinced that glacier mass balance (or rather glacier median elevation) can be a proxy to detect and monitor volcanic activity. Just because a process affects a variable doesn't mean that, conversely, this variable can be used to detect and study the process.

I challenge the authors to take the median elevations for all RGI glaciers and based on that, identify where volcanoes are active. This would be an entire new study so it would be unreasonable to ask authors to perform it at this stage of the review process.

The authors could have easily downplayed on the "volcano monitoring aspect" of their introduction and conclusion without affecting the scope of their study.

.

(Remarks on code availability)

Feeling the heat: globally, volcanoes stunt nearby glaciers

Response to referees

Tryggvi Unnsteinsson^{1*}, Matteo Spagnolo¹, Brice R. Rea¹, Tárilo Girona², Iestyn Barr³,
Donal Mullan⁴

¹School of Geosciences, University of Aberdeen, King’s College, Aberdeen, AB24 3FX, Scotland,
United Kingdom.

²Alaska Volcano Observatory, Geophysical Institute, University of Alaska Fairbanks, 2156 Koyukuk
Drive, Fairbanks, AK 99775, Alaska, USA.

³Department of Natural Sciences, Manchester Metropolitan University, Lower Ormond Street,
Manchester, M15 6BX, England, United Kingdom.

⁴School of Natural and Built Environment, Queen’s University Belfast, University Road, Belfast,
BT7 1NN, Northern Ireland, United Kingdom.

*Corresponding author(s). E-mail(s): t.unnsteinsson.23@abdn.ac.uk;

Below are point-by-point responses in blue text to each reviewer comment. We include any *“changed or added text from the revised manuscript in quoted blue italic text”* below relevant answers. The documents accompanying this “response to referees” file (r2r.pdf) are the revised manuscript (manuscript_revised.pdf), a file showing the differences between the initial and revised manuscripts (manuscript_difference.pdf), and a supplementary material file (supp_mat.pdf).

Comments of reviewer 1

Review of Unnsteinsson et al., “Feeling the heat: globally, volcanoes stunt nearby glaciers” Comments by Linda Sobolewski

This manuscript presents a very interesting study in which the authors analyze to what extent active volcanoes impact the mass balance of adjacent glaciers by looking at their relative median elevations in comparison to non-volcanically influenced glaciers. I think it is a very interesting approach to assess the global influence of volcanoes on

glaciers apart from disruption by major (single) events. The methods and data used in this study are comprehensible and well presented. Furthermore, the strengths and weaknesses of this work were identified and discussed.

The key results of this study are that volcanoes affect the mass balance of glaciers globally, even without having a major eruption. Relative median elevations increase with proximity to volcanoes (the closer the stronger the effect) with volcanic heat fluxes being identified as the main driving factor. Other effects were also taken into account and exceptions were presented. While I started reading the manuscript some questions were coming up, e.g.: How to exclude the effect of climate? Considering small scale characteristics and regional specifics? However, all these questions were answered in detail in a clearly understandable way and context was provided where necessary. Overall, I think this work presents important and also very interesting (and in the end surprising) results to better understand volcano-ice interactions in a global context. Although it is known that volcanoes often impact overlying or nearby glaciers—even in periods of quiescence, the percentage ($\sim 80\%$) is surprising.

The literature used in the manuscript is appropriate and includes very recent studies.

Although I think the work presented is very good and the manuscript is well written, I have some suggestions for improvement and I would like to see some changes of figures. After making these minor revisions I would recommend the manuscript for publication.

Specific comments

L45-56: I understand that the authors somehow have to provide an overview of what they have done and some context is needed. However, this part really reads like a methodology section. Is there a way to summarize this paragraph in a more general way and move the details to the methods section?

We have rewritten this paragraph to be more general and we have moved it into the Introduction. This should hopefully give a better overview of the terminology and processes being analysed at an appropriate place in the manuscript. We have further added referrals to the Methods section within the text and minimised repetitions.

“In this study we address the foregoing questions by looking at an easily-measurable proxy for glacier mass balance history: median glacier elevation. Median glacier elevations are comparable to equilibrium-line altitudes (ELAs) [10], a common descriptive metric of glacier mass balance [11]. Encoded in the median glacier elevation are the integrated effects over time of all components of the glacier mass balance: sub-, en-, and supra-glacial ablation and accumulation, along with ice flux. Greater accumulation and/or lesser ablation will drive advance of glaciers downslope, and conversely greater ablation and/or lesser accumulation will force glaciers to retreat to higher elevations. Differences in median elevation between nearby glaciers, where climatic forcing and consequent surface mass balances should be similar, can hence be examined to assess the potential effect of volcanism on glaciers [8, 12]. Observed effects on glacier elevation and hence mass balance are a snapshot of the long term adjustment of glaciers. The response time of glaciers to a mass balance forcing, e.g. volcanism, will depend on the scale of the forcing, climate, and glacier size, and could span multiple decades or longer [13]. Thus, any volcanic effect that we observe on glaciers will be

an integrated effect of all prior volcanic processes and events –such as volcanic eruptions, interactions with volcanic products, and volcanically-enhanced basal melting. Here, we analyse the relative difference in median elevations of all glaciers within a 5, 10, 20 and 40 km radius of the location of each of the world’s active Holocene volcanoes. We do this by utilising global datasets of glaciers and volcanoes: the 274 531 glaciers in the Randolph Glacier Inventory (RGI) [15] and the 1309 Holocene volcanoes from the database of the Smithsonian Institution’s Global Volcanism Program (GVP) [14]; thereof are a total of 40 667 glaciers located within 40 km of 307 active volcanoes.”

Section 2.4: I would consider having the “Conclusion” as a separate chapter since the caption for chapter 2 is “Results and discussion”.

The format for Nature Communications does not include a Conclusion section/chapter. We have changed this subsection name to “Conclusive remarks” to hopefully fit better with both the journal guidelines and the reviewer’s request.

One general comment regarding the figures: I know it’s not always easy to calculate the final size of figures in a manuscript, but I would try to roughly use the same size for legends and labeling of graphs. Sometimes the labels are very small and hard to read, even when zooming in. And sometimes they are disproportionately big compared to the actual figures.

We have now standardised as much of the figure making as we can for this manuscript. They should all be roughly equal but, as the reviewer points out, the final size may vary a bit depending on how the figures are typeset, which we will be able to further adjust once the manuscript proof is ready.

Figure 1a: I really like the concept of this one but it took a while before I recognized the blue lines in this figure (below the yellow one). Is there a possibility to play a bit with the colors and the contrast? Maybe make the background colors a bit lighter or more transparent to better highlight the relative median elevation lines?

We have changed the figure by making the background greyscale to better highlight the median elevation colour gradient.

If the “Median elevation” belongs to the inset graph (upper part of the figure) shouldn’t it be dashed?

We have changed this in the legend to be dashed and yellow to reflect the average median elevation.

Figure 1b: Please make the labels a bit bigger.

We have increased the size of the labels.

Figure 2: This might be a subjective opinion but would it be better not to make the dots transparent? It appears that you used more colors than red and blue since sometimes the dots look more orange and sometimes really red. I don’t think this will influence the key message of your figure although not every single dot will be visible. OR: did you use more colors to show a wider range? Then you should use a different color scale such as in Fig. 1a. Also, please make the legend a bit smaller and not hide data with it.

The transparency was initially intended to better display overlaid points, not to show a gradient. We have removed the transparency from the dots as suggested by the reviewer. We have also shrunk the legend as requested.

Were there any volcanoes where the MGE wasn't influenced at all?

Yes, there are volcanoes where we cannot observe any influence on MGE. These are discussed in the text:

“Two main reasons for glacier-on-volcano outliers are observed: (i) domination of non-volcanic factors on glacier mass balance; and (ii) poorly constrained volcano locations. ...”

Figure 4: It's super hard to distinguish between the different lines. Would it be possible to work with different colors or would it significantly influence the message of your figure when you zoom a bit more to the center and not show the full extent towards the upper and lower values? Or perhaps a 4a and 4b version where 4b is the version zoomed in to a scale that allows the reader to visualize the differences in the lines.

We have made the points gray to improve colour contrast, and we have “zoomed in”/made a cut-off at ± 1000 on the y-axis to help with visualisation.

Figure 5: I really understand the difficulty plotting so much data on one single map. Maybe plot individual areas separately to better visualize details? See for example Sobolewski et al. (2022), Figure 1 (<https://doi.org/10.1007/s00445-022-01525-z>)

We have both increased the resolution of the image to allow online readers to zoom into the figure, and we have added enlarged insets of four regions to the supplementary material. We opted to add the insets to the supp. mat., rather than making a very large figure with subfigures, as we find it difficult to justify such a large figure in the Methods section. We have added a reference to the new insets in the supp. mat., to the caption of Fig. 5:

*“...Enlarged insets of the areas with the highest concentration of glacierised volcanoes (dashed boxes **a-d**) are shown in Fig. S5.”*

Comments of reviewer 2

Review of Unnsteinsson et al., NatComm, March 2025

In their paper, Unnsteinsson et al. build on an earlier study by Howcutt et al (Geology, 2023, restricted to the Andes) to detect the influence that volcanic activity has at global scale on glacier health. This is done by intersecting two main database: a global inventory of glaciers and of actives volcanoes. The authors use the anomalies of the median glacier elevation (compared to a regional mean) to detect if volcanoes influence the glacier mass balance. A higher median glacier elevation is indicative of a shorter glacier and hence one for which there is an additional source of melt, not climate related. Authors conclude to a clear imprint of volcanic activities on glacier health.

This is an exciting study that, if this is confirmed, suggests a distinct influence of volcanic activity on glacier mass balance. I think it is appropriate for Nature Communication given the length and the fact that it elegantly intersects two fields of research, glaciology and volcanology. However, as explained below I am less convinced by the demonstration that this purely due to enhanced basal melting and also a proper definition of the criteria used to qualify “glacier health”. “Mass balance” is maybe not the right term, maybe even wrong...

We thank reviewer 2 for their comments that we believe highlight where we can improve phrasing and terminologies to avoid misunderstandings. Firstly, when we use the term “mass balance” we are referring to the overall mass balance of glaciers: a combined effect of sub-, en-, and supra-glacial melt and accumulation, along with the in- and out-flux of ice. In this study we investigate the effect volcanoes have on glacier mass balance, by using median elevation as a proxy for this, i.e., without having to directly quantify the mass balances. As specified below, we have rephrased parts of the text to hopefully make this explicit. Secondly, we have now made it clearer that enhanced basal melting, although likely to play the biggest role for the reasons explained in the main text when addressing other potential influences, is not the only driver of observed higher median glacier elevations around volcanoes. We have rephrased parts of the text, see response to comments **L19** and **L45-56**, to clarify that the global signal of volcanic effects on glacier mass balance is from a combination of volcano-induced melt processes.

General comments

1: Ref [9] cited in the introduction is a global dataset of individual glacier mass balance spanning the period 2000 to 2019 (sub-periods are also available but the longest period has the smallest uncertainties). Rather than relying on an indirect proxy of the mass balance (the median altitude, as also done in Howcutt et al.), authors could apply their methodology to this dataset. In my view, that could be an improvement of the paper and a strong confirmation that glacier are actually feeling the heat of volcanoes.

This is a general comment I wrote right away after reading the paper a first time and in particular the abstract where the word “glacier mass balance” is used several time.

Thinking about it, maybe I missed something and maybe the dataset in Ref [9] (i.e. their 20-yr glacier mass balance) is not the right one to detect the influence of volcanoes on glaciers. A reason for that would be that glaciers influenced by volcanoes already had time to build an hypsometry, different from their non volcanic neighbours (i.e. a different median altitude), such that the enhanced volcanic melt is “already” compensated by a location at higher reaches (where surface mass balance is more positive due to the mass balance gradient). If this is indeed the case, then the dataset in Ref [9] may not really help to detect the influence of volcanoes (backing up the choice by the authors).

After an initial speculation on the use of global mass-balance studies, such as Ref [9], we believe the reviewer correctly argues here that it would not help to detect the global effect of volcanism on glaciers. Ref [9] might possibly capture some short-term effects for individual volcanoes but, as identified by the reviewer, longer timescales would likely be needed for a global assessment. In essence, our study manages this by capturing everything that has come to pass prior to the collection of the RGI dataset.

These issues of response time, hypsometric adjustment and hence of the choice of the right variable to qualify the impact that volcanic activities have on glacier health is not discussed at all in the submitted manuscript and this is really missing.

We have added a paragraph to the introduction to discuss these processes in more detail. Please see text excerpt in response to comment **L45-56** of Reviewer 1.

In fact, if my reasoning above is true, it means that glaciers on volcanoes do NOT have mass balances different from others, but rather that they built an hypsometry that takes into account this additional source of melt. So the word “mass balance” may not be the right one to use.

As further clarified in the new introductory paragraph, we do not quantify glacier mass balance – we only observe what effect volcanoes had on glacier elevation, which has been demonstrated to be a good proxy for glacier mass balance, with the discussed caveat of it encapsulating a long-term response of glaciers to forcing.

2: Authors could also discuss at what time scale their results apply. Echoing a bit my first main comment just above, I guess this time scale is the time for the glacier to built an hypsometry where the volcanic effect is compensated by the distribution with altitude.

We have added a discussion of these processes in the introduction. Please see text excerpt in response to comment **L45-56** of Reviewer 1.

An interesting follow up question is how glaciers close/far from volcanoes will respond to ongoing warming. Do authors expect a similar or different response (assuming a similar climate forcings)? This should be touch upon in the article (not solved).

We do have a sentence in the Conclusion section addressing what our results could entail for glaciers near volcanoes in a warming climate:

“In a warming climate, this additional negative effect that volcanoes have on the mass balance of overlying glaciers could hasten their demise.”

A related question is the time-invariance of the volcanic activity. Again something to discuss, even briefly.

We have added a discussion of these processes in the introduction. Please see text excerpt in response to comment **L45-56** of Reviewer 1.

3: “though debris-enhanced melt may play a role for some volcanoes, it cannot explain the global scale of positive median elevation anomalies associated with volcanoes”. This is a very firm statement based on little evidences. I would be ready to accept such a statement if, for example, authors had examined some albedo maps during the months of minimum snow cover (from Modis) and confirm that the albedo is not statistically different for glaciers nearby volcanoes. The same applies for debris thickness for which global map of debris thickness could be used to verify that glaciers on volcanoes do not differ (see Herreid and Pellicciotti, 2020; Rounce et al., 2021). I understand this is a significant amount of work but otherwise the conclusion of “no debris influence” is not supported.

It was not our intention that readers would perceive our argument as that there was “no debris influence”. Debris can indeed have diverging effects on glaciers, enhancing or inhibiting ablation, depending on the debris thickness and composition. Our reasoning is rather that due to this diverging effect, along with the lack of eruptions for the majority of glacierised volcanoes, debris presence cannot explain the global signal of high median elevation of glaciers on volcanoes. Hence, our emphasis on volcanic heat. We believe that adding a global study of albedo and debris thickness, if feasible, would unlikely improve on existing global estimates unless informed by detailed in-situ measurements, which do not exist on a global scale. The combined effect of debris will inevitably be site specific, but as an example, the regional estimate of Rounce et al.(2021)[19] for Iceland (where the largest glaciers are in close proximity to volcanoes) show a very small (<5 %) debris coverage of glaciers which result in the largest relative reduction of surface melt of all regions.

To address this comment, we have lessened our statements and clarified the text in the second paragraph of section 2.2 to make sure that the role of debris-enhanced melt is not ruled out. We have added to the discussion of the diverging effects of debris on glaciers, and we added a discussion of the immediate, intermediate, and long-term effects of erupted tephra on glacier mass balance, accompanied by more citations to relevant papers. The rewritten paragraph is included here below: (changes to it may be better seen in the difference document L114-140)

“There are likely three main controlling factors pertaining to volcanoes that could influence the relative elevation of nearby and overlying glaciers: presence and production of debris; prominence above surrounding topography; and volcanic heat fluxes. Volcanoes generally produce high fluxes of sediments from their flanks [16], and can be subject to heightened levels of erosion by the activity of overlying glaciers [17]. This, in conjunction with volcanic ejecta, may enhance debris entrainment and covering of glaciers proximal, or down wind of, volcanoes. As debris covers thicken they rapidly begin to shield glaciers from ablation [18, 19], resulting in lower median glacier elevations. A visual inspection of our results confirms that thicker and more expansive debris coverage is indeed linked to lower median elevations of glaciers both near and away from volcanoes (e.g., dark blue glaciers in Fig. 1b, S2, and S4). Hence, debris-shielding cannot probably drive higher median elevations around volcanoes. Conversely, impurities in

ice or sufficiently thin debris cover, e.g., tephra from eruptions, can enhance ablation by decreasing glacier surface albedo and increasing solar energy absorption [20, 21]. This could result in higher median glacier elevations. Debris-enhanced melt could be particularly prevalent following volcanic eruptions that blanket glaciers with tephra [21]. Though, glaciers can also be positively affected if the tephra layer is thick enough [20, 22]. The longer-term effect of erupted tephra on glacier ablation is complex and site specific. In the years immediately following an eruption, tephra may both: continue lowering the albedo of glaciers, potentially increasing ablation, before rebounding to previous levels [21, 23]; or carry on shielding and reducing ablation rates, leading to glacier preservation and potential advance [24, 25]. Subsequently, reworking and redistribution of tephra on glaciers can also both promote and inhibit ablation depending on glacier and surface characteristics [25, 26]. Thus, due to the potentially diverging effects tephra can have on glaciers, tephra-enhanced melting is unlikely to alone explain the global signal of higher median glacier elevations. This is further supported, as only around 30-35% of glacierised volcanoes had confirmed eruptions in the last century [14].”

Specific comments (including some substantial comments).

L19: It is a bit hard to grasp the effect said this way. "lead to more negative mass balances" would be clearer I think. My main comment 1 above may lead to a rewriting of this and not using “mass balance”. The mass balance of volcanoes-influenced glaciers may NOT be different now because glaciers already had time to adjust their hypsometry to difference in melt.

It is true that this could possibly be worded better. However, we must be careful to not state that the total mass balance is negative – but only that volcanic processes have a negative impact. It could possibly be that some volcano-influenced glaciers have adjusted their hypsometry (by retreating to higher elevations), where their total mass balance reaches an equilibrium. Observations of glaciers with changed hypsometry (captured by a higher median elevation) would then in turn signal that the underlying mass balance is being negatively effected. We have rephrased this sentence in the text to hopefully better reflect this:

“We show that glaciers near volcanoes are progressively confined to higher elevations closer to volcanoes, suggesting that volcanic processes have a negative impact on nearby glacier mass balance.”

L20: Are the "adjacent glaciers" mentioned above within this distance of the volcanos? (I guess not but the abstract need to define what an "adjacent glacier" is)

We have rephrased the sentence mentioning “adjacent glaciers” to specify the distance from the volcanoes:

“In this study, we analyse to what extent 307 active volcanoes worldwide impact the mass balance of glaciers within a 40 km radius, a total of 40 667 glaciers, by comparing their relative median elevations.”

L45-56: Authors need to refer to the methods section in the “main”

Same response as to comment **L45-56** of Reviewer 1. We have rewritten this paragraph to be more general and we have moved it into the Introduction. This should hopefully give a better overview of the terminology and processes

being analysed at an appropriate place in the manuscript. We have further added referrals to the Methods section within the text in the Results section to not be repeating the methodology.

L49: Radii around what center? Volcanoes? Make it clear and tell the readers how the volcanoes location is obtained. How two nearby volcanoes are treated (to avoid redundancy), as many volcanoes will be clustered. This comment will be easy to answer by referring more specifically to the Methods section (I did not know this Methods existed when I read the article first)

We have fixed the text in the Methods to explicitly state how the distance is computed, and how the locations of volcanoes are taken:

“As a descriptive distance between volcanoes and glaciers, we use the minimum distance between volcano locations, given in Ref. [14], and glacier outlines, with zero indicating that a volcano is directly overlain by a glacier.”

L51: define the related time period (in years)

We have added a definition of the time period in years to the text:

“...each of the world’s active Holocene volcanoes (erupted in the past 12 000 years [14]).”

L56: Median altitude is an indirect proxy of the mass balance and rather reflect the mass balance history. This needs to be clarified (see general comment 1).

We have added a clarification into the new paragraph at the end of the Introduction. Please see text excerpt in response to comment **L45-56** of Reviewer 1.

Figure 1a: the blueish colours for the median elevations are hard to distinguished from the background.

We have changed the figure by making the background greyscale to better highlight the median elevation colour gradient.

L68: Authors could annotate panel 1b to make sure this is clear to all readers which glaciers are concerned.

We have added annotations to Fig 1b, and now refer to them in the figure caption:

“...The dashed boxes are to highlight the debris-covered glaciers and dry-calving glaciers of Sanford (1) and Drum (2).”

L82: I did not understand this statement. The first part of the sentence is about number of glaciers in the vicinity and the second part of about glacier types.

To clarify these statements we have split the sentence into two, and clarified further:

“ For statistical significance, we limited the analysis to volcanoes with at least four nearby or overlying glaciers. Despite not filtering out, or adjusting for, different glacier types or characteristics, the positive anomalies are still observed globally in all statistical tests (Tab. 1 and Methods).”

Table 1: Are the percentages given for the number of volcanoes for which the increase in median glacier elevation is statistically significant? Or for all, i.e. without considering the statistical significance? My understanding is that the authors compared three different ways to compute the trend but when did they accounted for the significance of the trend? Maybe I missed something but this table needs to be clarified.

The percentages are given for all volcanoes. We have amended the caption to better reflect that:

“The total number, N , of volcanoes and glaciers within a given radius and the percentages, P , of all the volcanoes, with at least four glaciers, that show an increase in median glacier elevations towards them.”

L93: I find it strange to cite an article in the "Arabian Journal of Geosciences" to support the impact of debris on glacier melt. I can think of this study, looking in detail at the albedo in Iceland but there must be many others

Gunnarsson, A., Gardarsson, S. M., Pálsson, F., Jóhannesson, T., and Sveinsson, Ó. G. B.: Annual and inter-annual variability and trends of albedo of Icelandic glaciers, *The Cryosphere*, 15, 547–570, <https://doi.org/10.5194/tc-15-547-2021>, 2021.

We have added a citation to Gunnarsson et al. (2021), but we keep Dragosics et al. (2016) as we find it relevant to the discussion. It is worth noting that Gunnarsson et al. (2021) themselves cite Dragosics et al. (2016). Please see text excerpt in response to general comment 3.

L94: Herreid and Pellicciotti provide a global debris thickness map on glacier that authors could use to test the hypothesis (see general comment).

Please see response to general comment 3.

L98: The statement of few tephra-generating eruptions in the last century needs to be backed up by a reference.

We have amended the sentence to better reflect that the argument is on the number of eruptions, not specifically tephra-producing eruptions. We have further added a citation to the Smithsonian’s Global Volcanism Program database where the eruption data is from:

“Debris-enhanced melt could be particularly prevalent following volcanic eruptions that blanket glaciers with tephra [21]. However, post-eruptive tephra-enhanced melting cannot alone explain the global signal of higher median glacier elevations, as only around 30-35% of glacierised volcanoes had confirmed eruptions in the last century [14].”

L106: Authors need to show some plots/statistics for Alaska Range, St Elias Mountains, Alps, and High-mountain Asia (in a supplement?) in order to back up the comparable median glacier elevations whatever the maximum height of the summit. Really important.

We have added figures for these areas and added to a supplementary file, and now refer to these figures in the text:

“This is the case in our results for glaciers near other non-volcanic prominent peaks around the world, e.g., in the Alaska Range, St Elias Mountains, Alps, and High-Mountain Asia (supplementary Figs. S1, S2, S3, and S4).”

L112: “The only plausible driver of higher median elevations for glaciers located on or around volcanoes. . .”. "Only" can be used if the effect of debris (albedo) can be ruled out. This has not been demonstrated yet (see general comments).

We were not distinguishing between melt processes here, but we have amended these sentences to better clarify and avoid misunderstandings:

“Given the caveats associated with the two other main drivers discussed above, the most plausible driver of higher median elevations for glaciers located on or around volcanoes, in a global context, would seem to be volcanically-induced basal melt. This melt could be attributed to any volcanic processes during both quiescent and active periods, and either be localised or distributed.”

That is, if there is debris-enhanced melt (due to lower albedo) on volcanoes – then that too is volcanically induced melt.

Figure 3: Do authors have an explanation why the correlation coefficient seems to agree rather well between LR and SR, but MK seems to lead to less negative values? This is discussed a bit in the Methods section but could be also briefly mentioned in the main.

We have added a brief mention of this in the main body of the manuscript:

“Despite not filtering out, or adjusting for, different glacier types or characteristics, the positive anomalies are still observed globally in all statistical tests (Tab. 1 and Methods). The Mann-Kendall test shows weaker correlation than the other tests for larger radii (Tab. 1), which is expected as it does not directly consider the distance from the volcanoes (Methods). The considerable agreement between the tests demonstrates...”

L125: I did not understand why an enhanced mass turnover would negatively impact the glacier mass balance. Can the authors cite a reference to back up this statement and provide more explanations? Use the term mass balance carefully...

“Mass turnover” was perhaps a poor phrasing and we have changed it to “ice flux” in the text:

“Short periods of increased velocities could elongate glaciers, but in the long term, sustained faster velocities would likely yield an accelerated ice flux, and thus negatively impact glacier mass balance leading to glacier shrinking and retreat.”

As previously mentioned we are referring to the total mass balance of glaciers, not just the surface mass balance. The statement that enhanced ice flux negatively impacts glacier mass balance is a direct result of the continuity equation (e.g., Eq 5.3 in [13] and Eq 8.69 in [Cuffey & Paterson, 2010 - The Physics of Glaciers]):

$$\frac{\partial H}{\partial t} = -\nabla \cdot \dot{Q} + \dot{b},$$

where H is ice thickness, \dot{Q} is ice flux, and \dot{b} is combined basal-, englacial-, and surface mass balance (melt and accumulation). For a constant accumulation and melt, an increase in ice-flux will hence negatively affect the overall mass balance – the glacier will thin and evacuate more ice towards the ablation zone.

L180: Make it clear that you use the RGI attributes directly and did not recompute them. (if I understood well)

We have amended the sentence to explicitly state that we use the RGI attributes

“The topographic attributes of glaciers used in this study are already included in the RGI v7.0 dataset [15]. These topographic attributes...”

L189: Authors need to clarify how the distance is computed. Do they find the distance to the centroid of the polygon?
Or the distance to the closest segment of the polygon?

We have changed the word “polygons” to “outlines” to indicate that the distance is computed from the closest segment of the glacier polygons:

“As a descriptive distance between volcanoes and glaciers, we use the minimum distance between volcano locations, given in Ref. [14], and glacier outlines, with zero indicating that a volcano is directly overlain by a glacier.”

Feeling the heat: globally, volcanoes stunt nearby glaciers

Second response to referees

Tryggvi Unnsteinsson^{1*}, Matteo Spagnolo¹, Brice R. Rea¹, Tárilo Girona², Iestyn Barr³,
Donal Mullan⁴

¹School of Geosciences, University of Aberdeen, King's College, Aberdeen, AB24 3FX, Scotland,
United Kingdom.

²Alaska Volcano Observatory, Geophysical Institute, University of Alaska Fairbanks, 2156 Koyukuk
Drive, Fairbanks, AK 99775, Alaska, USA.

³Department of Natural Sciences, Manchester Metropolitan University, Lower Ormond Street,
Manchester, M15 6BX, England, United Kingdom.

⁴School of Natural and Built Environment, Queen's University Belfast, University Road, Belfast,
BT7 1NN, Northern Ireland, United Kingdom.

*Corresponding author(s). E-mail(s): t.unnsteinsson.23@abdn.ac.uk;

Below are point-by-point responses in blue text to each reviewer comment. We include any *“changed or added text from the revised manuscript in quoted blue italic text”* below relevant answers. The documents accompanying this “response to referees” file (r2r_02.pdf) are the revised manuscript (manuscript_revised_02.pdf), a file showing the differences between the initial and revised manuscripts (manuscript_difference_02.pdf), and a supplementary material file (supp_mat.pdf).

Comments of reviewer 1

The authors have invested a lot of work to revise the manuscript and address all comments and concerns. I think it is appropriate for Nature Communication in its current form and I would recommend the manuscript for publication.

Comments of reviewer 2

Review of the revised version of Unnsteinsson et al., NatComm, June 2025

The article has been improved, taking into account most comments from the original reviews. However, I cannot recommend publication yet because there is still some ambiguity in terminology: median glacier elevation (studied here) and mass balance need to be more clearly distinguished. I tried to highlight all places where some improvements are needed.

Abstract: the background justification of the work is not appropriate. As if an analysis of the impact of volcanoes on glacier health could be a way to "monitor" volcanoes. This would be really indirect because of many other processes influencing glaciers (including climate change) and because the time (at least decadal) it takes for a glacier to adjust its hypsometry to a change in volcanic heat. Hence, the first sentences of the abstract need to be rewritten.

We must stress that we employ the use of elevation metrics, as proxy of glacier mass balance, to study the effect of volcanoes on glaciers in a global context. By doing so, we have demonstrated the existence of a measurable volcanic effect on glaciers around the majority of glacierised volcanoes, which indicates the undeniable potential of using glacier mass balance studies, in whichever form, as an additional tool for volcano monitoring. Indeed, localised glacier mass balance studies have been commonly used to monitor underlying volcanic activity, e.g., through the monitoring of ice cauldrons and other melt features [see citations in Introduction]. Volcanic processes can operate over long timescales [e.g., Girona et al., 2021, <https://doi.org/10.1038/s41561-021-00705-4>] and thus long-term analysis of glacier change could represent an additional precursor to take into consideration. As with many other monitored precursors of glacierised volcanoes, e.g., deformation, signals may be complicated by various processes, e.g., seasonal loading and unloading of snow and ice, isostatic rebound due to climate change, poor image correlations due to snow/ice cover, and so forth.

L37-38: unclear what is meant. "local to" seems isolated. "metrics" is vague.

Sentence rephrased to clarify:

"...or inferred it by comparing elevations of debris-free and land-terminating glaciers, $\geq 1 \text{ km}^2$, within 15 km of volcanoes [8]."

L45: "Median glacier elevations are comparable to equilibrium-line altitudes (ELAs), a common descriptive metric of glacier mass balance" Need to specify the time scale at which this applied. For example inter-annual variability of the glacier median elevation is very small (glacier geometries change slowly) whereas the mass balance inter-annual variability is large (because of the climate variability). So this statement only applies to long time scales (century long time scale) and this needs to be explained.

Sentence has been rephrased to clarify that we are not discussing the seasonal ELAs but the theoretical steady-state ELAs:

"Median glacier elevations are comparable to theoretical steady-state equilibrium-line altitudes (ELAs)..."

Title of section 2.1: Authors did not detect "mass balance anomalies". But different median glacier elevations. Hence, this section title is misleading.

Title has been changed to: "Detection of anomalous median glacier elevations"

L93-94: See for example the factors listed in Huss et al., 2012 : <https://tc.copernicus.org/articles/6/713/2012/>

Sentence has been rephrased to name a couple of examples, along with a citation instead of exhaustively list all factors influencing glacier mass balance:

"The considerable agreement between the tests demonstrates that the signal is strong enough to offset the potential effect of other factors that are known to affect glacier mass balance [13], e.g., aspect and distance from moisture source, which might be able to mask the impact that volcanoes have on glaciers."

Fig 4: I suggest using another color than red for the dashed fit to really make the curves different (hard to see the dashed one as it overlays a black line)

We have changed the red dashed line to a whole orange line, and increased the line width to help with visualisation.

L127-128: Maybe I missed something but there seems to be a contradiction with the previous sentence. If the max glacier elevation increase and the minimum elevation stay constant then the range increase and the length also.

We have added a sentence in between these two sentences to better clarify that these glaciers are indeed getting longer, but not as long as expected:

"Instead, for glaciers near and overlying volcanoes we observe that as relative maximum glacier elevations increase nearer to volcanoes, the relative minimum glacier elevations stay relatively constant (Fig. 4). Whereas theory would suggest that higher maximum elevations should correlate with lower minimum elevations. That is, glaciers on or near volcanoes have a reduced elevation range, and hence shorter length than expected for glaciers sitting on topographically prominent peaks."

Title 2.3: replace "mass balance" by "median elevation"

Title rephrased to: "Footprint of volcano-induced glacier median elevation anomalies"

L174: I do not think mass balance can be a proxy of volcanic activity.

Should the issue be the use of "proxy" in this context, then we agree and have changed the phrasing from "using glacier mass balance as a proxy for volcanic activity" to "using glacier mass balance to infer on volcanic activity."

However, if the reviewer does not think glacier mass balance can be used to infer on volcanic activity, then we must respectfully disagree. Previous work has shown the vastly differing volcano/geothermally induced melt rates between glaciers near and distal to volcanoes in Iceland [7], and hence clearly demonstrate the potential of using glacier mass balance studies to infer on volcanic activity.

As said in my original review and repeated here, the RGI glaciers already adjusted their hypsometry (as actually demonstrated by this study) to the influence of volcanoes by standing at higher median elevations.

The reviewer accepts our results that glacier have adapted their hypsometry to previous volcanic activity. Based on our results, it can only be plausible, and indeed likely, that glaciers will continue to adjust their hypsometry in

response to any on-going and future volcanic activity. This is not to say that there will be no other influencing factors on glacier mass balance complicating the picture, nor that glacier mass balance alone should be used as a monitoring tool, or that the response of any glacier to volcanic forcing will be rapid. However, our results demonstrate a strong link between volcanic forcing and glacier median elevation. Hence it is only plausible that long term change in volcanic forcing will translate into a measurable adjustment of glacier hypsometry. Such mass balance studies would likely not give insight into rapidly evolving scenarios, such as volcanic eruptions, but are more likely to inform on longer-term volcanic trends. We have amended the sentence to highlight this long-term aspect:

“ Our results highlight where future long-term monitoring efforts of glacierised volcanoes...”

Hence, their present-day mass balance is very likely not different to their neighbours.

The surface mass balance, dictated by climate, is indeed likely not very different between the neighbouring glaciers. However, the volcanic effect on total mass balance can be significant between glaciers – see values reported in Ref. [7].

[7] Jóhannesson T. et al. Non-surface mass balance of glaciers in Iceland. *Journal of Glaciology* 66(258):685-697 (2020). <https://doi.org/10.1017/jog.2020.37>

Fig S1-S4: The number for the summits are so clustered that there are hard to distinguished. Sometime the numbers (in black) are also hard to read because of the dark background colors. These figures need some edits.

Mountain labels have been moved inside their markers (white numbers on black triangles) to improve visualisation. Figures are also provided in high resolution to allow readers to zoom in (peer-review figures had to be downsampled due to upload limits).